# Complementary peptides represent a credible alternative to agrochemicals by activating translation of targeted proteins

Mélanie Ormancey [1,11], Bruno Guillotin [1,2,11], Rémy Merret [3], Laurent Camborde[1], Carine Duboé[1], Bertrand Fabre[1], Cécile Pouzet [4], Francis Impens [5,6,7], Delphi Van Haver [5,6,7], Marie-Christine Carpentier[3], Hélène San Clemente[1], Marielle Aguilar[1], Dominique Lauressergues[1,2], Lars B. Scharff [8], Carole Pichereaux [4,9,10], Odile Burlet-Schiltz[9,10], Cécile Bousquet-Antonelli [3], Kris Gevaert [5,7], Patrice Thuleau[1], Serge Plaza[1] & Jean-Philippe Combier [1,2] ✉

The current agriculture main challenge is to maintain food production while facing multiple threats such as increasing world population, temperature increase, lack of agrochemicals due to health issues and uprising of weeds resistant to herbicides. Developing novel, alternative, and safe methods is hence of paramount importance. Here, we show that complementary peptides (cPEPs) from any gene can be designed to target specifically plant coding genes. External application of synthetic peptides increases the abundance of the targeted protein, leading to related phenotypes. Moreover, we provide evidence that cPEPs can be powerful tools in agronomy to improve plant traits, such as growth, resistance to pathogen or heat stress, without the needs of genetic approaches. Finally, by combining their activity they can also be used to reduce weed growth.

Chemicals are extensively used in modern agriculture. Together with agricultural mechanization, genetics, and crop management, chemicals have been primarily responsible for the huge improvement in crop yields since 1945 (http://www.kingcorn.org/news/timeless/YieldTrends.html). Chemicals fight against weeds or pathogens or act as growth regulators, such as hormones and fertilizers[1].

World agriculture is facing huge challenges in the coming years to feed the increasing world population[2]: the increasing temperature will decrease crop yields, and the lack of mineral phosphorus-based fertilizers, which are limited resources extracted from soils, will be detrimental to worldwide agriculture. In addition, the use of chemicals in today's agriculture faces two major problems: first, more and more weeds/pathogens are becoming resistant to currently used pesticides[3]. The second issue is social acceptance and growing public concern about molecules polluting soils and/or are dangerous for animal and human health[4]. In this context, identifying safe and natural molecules increasing crop yields is one of the biggest challenges that plant biologists have to face.

We present here the identification of peptides capable of modulating the expression of protein-coding genes, simply by their external application. Furthermore, we show that these peptides can be used in agronomy to improve crop development while decreasing weed growth, and that certain phenotypes that are difficult to manage with

[1]Laboratoire de Recherche en Sciences Végétales, CNRS/UT3/INPT, Auzeville-Tolosane, France. [2]Micropep Technologies, Auzeville-Tolosane, France. [3]Laboratoire Génome et Développement des Plantes, CNRS/UPVD, Perpignan, France. [4]Fédération de Recherche Agrobiosciences, Interactions et Biodiversité, UT3/CNRS, Auzeville-Tolosane, France. [5]VIB Center for Medical Biotechnology, B9052 Ghent, Belgium. [6]VIB Proteomics Core, B9052 Ghent, Belgium. [7]Ghent University, Department of Biomolecular Medicine, B9052 Ghent, Belgium. [8]University of Copenhagen, Department of Plant and Environmental Sciences, Copenhagen, Denmark. [9]Institut de Pharmacologie et de Biologie Structurale, CNRS/UT3, Toulouse, France. [10]Infrastructure nationale de protéomique, ProFI, FR 2048 Toulouse, France. [11]These authors contributed equally: Mélanie Ormancey, Bruno Guillotin. ✉e-mail: jean-philippe.combier@univ-tlse3.fr

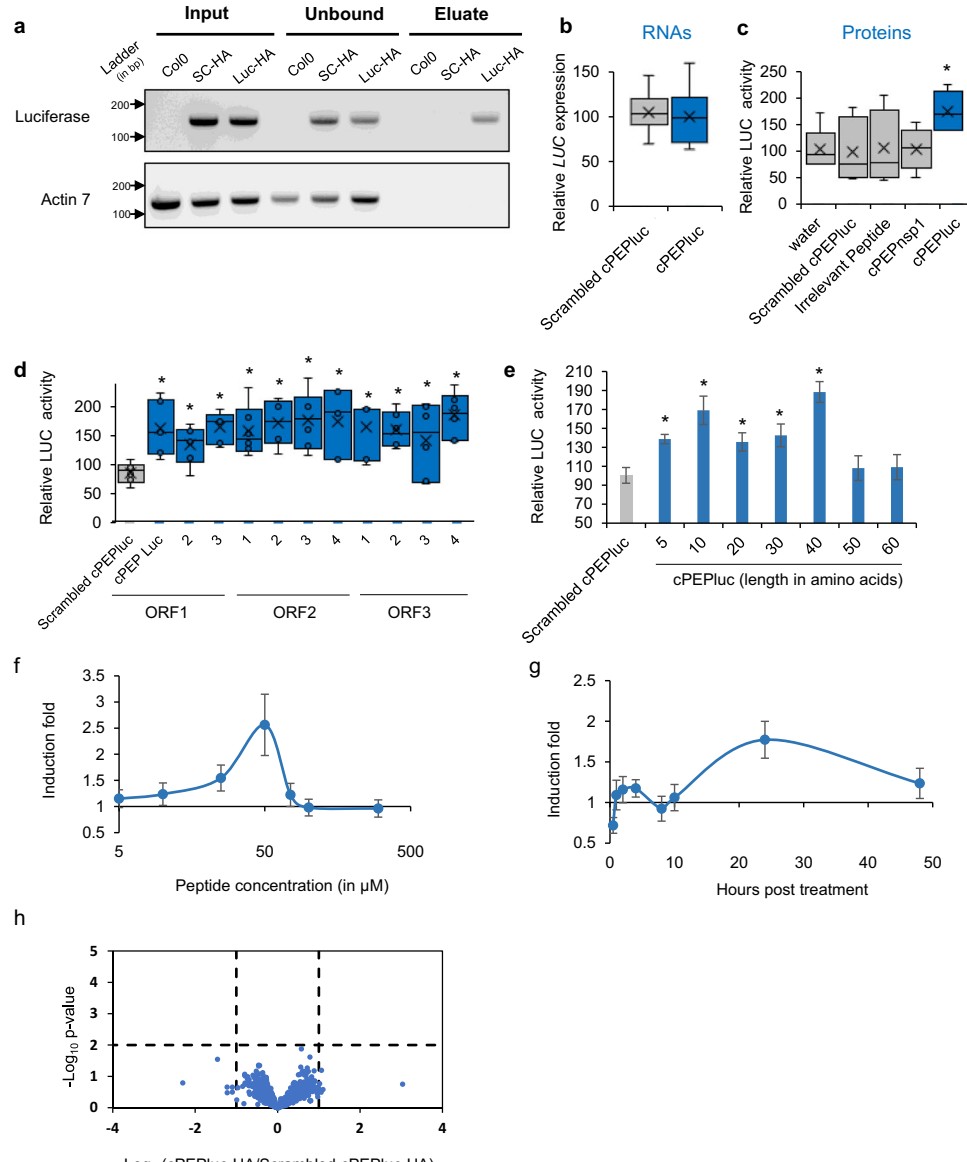

**Fig. 1 | Complementary peptides increase protein expression.** Analysis of luciferase expression in *A. thaliana* plants constitutively expressing *LUC* transgene. **a** Agarose gel after PCR amplification of RNA immunoprecipitation using anti-HA magnetic beads followed by RT-PCR, on plants treated with the indicated peptide (SC-HA: Scrambled cPEPluc-HA; Luc-HA: cPEPluc-HA). **b** Relative expression of luciferase transgene in plants treated with the indicated peptide, quantified by RT-qPCR. **c**−**e** Relative LUC quantification of plants treated with the indicated peptide. **f**, **g** Ratio of induction of LUC quantification of plants treated with cPEPluc compared to plants treated with a scrambled cPEPluc, using various concentrations of peptides (**f**) or harvested at a different time point (**g**). **h** Proteomic analysis of plants treated with cPEPluc-HA compared to plants treated with scrambled cPEPluc-HA. For box plots, the cross represents the mean, the line shows the median value, and the upper part and lower part of the box represent the first and third quartiles. The error bars represent the minimal and maximal value for box plots and the standard error of the mean (SEM) for others. The asterisks indicate significant differences between the test and the control (*p* < 0.05). Asterisks indicate a significant difference between the test condition and the control according to the Wilcoxon test (**a**, **h** *n* = 5; **b**−**d** *n* = 6; **e** *n* = 8; **f**, **g** *n* = 12; *p* < 0.05). Source data are provided as a Source Data file.

chemicals, such as heat resistance and chlorophyll content, can be modulated by these peptides.

## Results

### Complementary peptides interact with their corresponding RNA

It has been previously shown that short peptides, produced *in planta*, can physically interact with their nascent RNA sequences[5]. We extended this concept by questioning whether any other short peptide could also interact with its mRNA. We designed a hydrophilic 10-amino acid-complementary peptide (cPEP, Supplementary Fig. 1 and Supplementary Data 1), corresponding to a ten-amino acids fragment of

the luciferase protein, constitutively expressed in *Arabidopsis thaliana*[6], and carrying no homology with any sequence of the *A. thaliana* genome. Using the RNA IP method followed by PCR on plants treated with HA-tagged peptides, we validated that such a cPEP was able to specifically interact with luciferase mRNA, while a corresponding scrambled peptide could not (Fig. 1a).

### cPEPs increase the activity of their targeted protein

To determine whether this interaction might have biological relevance, we treated plants with the synthetic cPEPluc, and analyzed luciferase expression. Whereas qPCR analysis did not show any effect on mRNA abundance (Fig. 1b), treatment with the peptide increased

the luciferase activity compared to treatments with water, scrambled cPEPluc, or luciferase-nonspecific peptides (Fig. 1c). We then designed ten additional 10-amino acids peptides targeting distinct sequences of the luciferase gene, chosen without a priori, in the three frames (ORF1 corresponds to the canonical luciferase protein, Supplementary Fig. 1), and all were able to increase luciferase activity (Fig. 1d). In parallel, we designed and synthesized 7 cPEPs, from 5 to 60 amino acids (Supplementary Fig. 1 and Supplementary Data 1). Exogenous treatment with these different peptides revealed an increase in luciferase activity for peptides ranging from 5 to 40 amino acids, whereas longer peptides had no effect (Fig. 1e). We then analyzed the effect of peptide concentration and time of treatment on the cPEPluc activity. The optimal concentration of cPEP in these conditions was found to be 50 μM (Fig. 1f) and 24 h of treatment showed the maximal effect of cPEP (Fig. 1g). Finally, to know whether cPEPs might have side effects, we treated *A. thaliana* plants expressing luciferase with a cPEPluc or its scrambled peptide and performed a proteomic analysis (Fig. 1h). Interestingly, we were not able to detect any significant changes ($p < 0.01$) in the proteome of plants treated with the cPEPluc compared to plants treated with the scrambled peptide, suggesting a strong specificity of cPEPs for their target protein.

### RNA corresponding to cPEP is required for cPEP activity

We next targeted another protein in another plant species: the *Medicago truncatula nodulation signaling pathway 1* (*NSP1*) protein, which is a GRAS transcription factor involved in root development[7]. We expressed *NSP1* mRNA in *Nicotiana benthamiana* leaves by agroinfiltration and used FRET-FLIM to assess the interaction between SYTOX-labeled *NSP1*-mRNAs and the exogenously applied FAM-tagged 10-amino acid-peptide cPEPnsp1. The results revealed close proximity, in the cytoplasm, between *NSP1* mRNA and the peptide, which was canceled when the mRNA sequence corresponding to the peptide was removed in *NSP1* mRNA (*NSP1 ΔcPEP*, Fig. 2a, and Supplementary Table 1). To know whether this interaction might have biological relevance, we treated *M. truncatula* plants with the synthetic cPEPnsp1, and analyzed NSP1 expression, at the mRNA and protein levels. Whereas qPCR analysis did not show an effect on mRNA abundance (Fig. 2b), treatment with the peptide increased the quantity of NSP1 proteins, as revealed by a translational fusion between *NSP1* and the *GUS* gene (Fig. 2c). Consistently with the FRET-FLIM results, deletion of the *NSP1* RNA sequence corresponding to the peptide canceled the effect of the peptide on protein abundance (Fig. 2d). In addition, when we replaced in *NSP1* this sequence by a random sequence (a construct we named *NSP1 ΔcPEP-Irrelevant Peptide*), we observed a gain of function for the corresponding irrelevant peptide, which was then able to increase NSP1 protein expression (Fig. 2e).

### cPEPs can target most proteins

To investigate whether cPEPs can target any protein, we studied and validated the activity of 11 cPEPs targeting 11 different proteins for which antibodies were available, in three different plant species (Supplementary Table 2). Finally, we wondered if the increased amount of protein observed after cPEP treatment was sufficient to modulate plant development. To address this, we first focused on different proteins of the model plant *M. truncatula*. Application of cPEPnsp1 decreased the quantity of lateral roots (Fig. 2f, g), while *nsp1* mutant and overexpression of NSP1 remained insensitive to the peptide (Fig. 2g). The *M. truncatula Sickle* (*SKL*) gene is an ortholog of *A. thaliana AtEIN2*, and *skl* mutants develop more nodules than wild-type plants[8]. Consistently, the application of a cPEPskl led to a decrease in the quantity of nodules (Fig. 2h, i). We then targeted the *M. truncatula* RH10 protein, which positively modulates plant defense against the pathogenic oomycete *Aphanomyces euteiches*[9]. Plant treatments with a cPEP targeting MtRH10 increased plant resistance to the pathogen, as

revealed by increased root development and decreased α-tubulin expression of *A. euteiches* in roots (Fig. 2j, k).

### cPEPs can be used to modulate different phenotypes

We next focused on different *A. thaliana* proteins involved in different plant functions. Firstly, treatment of *A. thaliana* seedlings with a cPEPdcl1 led to a decrease in primary root growth, which is coherent with the fact that *dcl1* mutants show longer main root[10] (Fig. 3a, b). We next treated *A. thaliana* plants with cPEPs targeting regulators of chlorophyll content. Interestingly, we were able to identify one cPEP, targeting the ABI5 protein, decreasing chlorophyll content, and one cPEP targeting the SGR1 protein, increasing this content (Fig. 3c). Similarly, a cPEP targeting HSP101, a protein involved in heat stress tolerance[11], improved seedling viability to heat shock (Fig. 3d). In parallel, we targeted several *A. thaliana* plant defense regulators. Interestingly, these cPEPs enhanced plant defense against the necrotrophic fungus *Botrytis cinerea*, as revealed by a decreased lesion size observed in plants treated with cPEPs compared to plants treated with an irrelevant peptide (Fig. 3e). We then used the cPEP with the strongest effect, cPEPcpk3, and we performed the same experiment using in parallel the *A. thaliana cpk3* mutant. Consistently, the cpk3 mutant did not respond to the cPEP (Supplementary Fig. 2). We then designed several cPEPs targeting different proteins involved in plant development and measured the flowering day of *A. thaliana* plants. We were able to identify cPEPs accelerating flowering (SHY2 and MRB1), while others were able to decrease plant development (BRI1, BAK1, TAP46, SPT, EIN2, GA2OX7, PHYB, HAG5, SHR, and WUS; Fig. 3f). Finally, we investigated whether cPEPs could have additive effects, by mixing some of them. Interestingly, when each peptide separately decreased development by up to 17% (Fig. 3f), a mixture of cPEPs targeting EIN2, BRI1, BAK1, and WUS, decreased development by 23%, as revealed by flowering day (Fig. 3g, h) and leaf growth (Fig. 3i, j), showing an additive effect of cPEPs. All these data show that, along with enhancing protein expression, cPEPs can be useful tools to precisely modulate several plant phenotypes.

### cPEPs activate the translation of their targeted protein

cPEPs increase the quantity of proteins without disturbing mRNA levels, suggesting that cPEPs increase protein translation or stability. In order to distinguish between both, we treated *A. thaliana* plants expressing the *LUC* gene with cPEPluc and cycloheximide (CHX), a translation inhibitor. A luciferase activity assay showed that the effect of cPEPluc was inhibited in the presence of CHX, implying that cPEPs do not act on protein stability but rather on protein translation (Fig. 4a). To support these data, we expressed the *LUC* gene with or without cPEPluc in wheat germ in vitro transcription/translation system, where no protease activity occurs. This revealed that cPEPluc increased LUC activity in vitro, strongly reinforcing the idea that cPEPs increase the efficiency of protein translation (Fig. 4b).

### cPEPs interact with ribosomal machinery

Up-regulation of translation could be the consequence of an increase in ribosome recruitment or a faster elongation rate. To discriminate between both possibilities, profiling of mRNA degradation through 5′P Sequencing (5′P-Seq) has been used to assess ribosome accumulation along a transcript at defined positions[12]. Analysis of ribosome occupancy on CPK3 mRNAs showed increased ribosome accumulation upstream and downstream of the start codon in cPEPcpk3 treated samples (Fig. 4c and Supplementary Fig. 3a). These data reveal that cPEP treatment increases ribosome recruitment at translation initiation sites specifically on CPK3 transcripts but not on other transcripts, such as CPK6, CPK9 or CPK32, which are close homologs to CPK3 (Supplementary Fig. 3b). Finally, we asked how cPEPs can activate translation of their target transcript. To achieve this, we treated plants with an HA-tagged cPEPcpk3 (cPEPcpk3-HA) or its corresponding

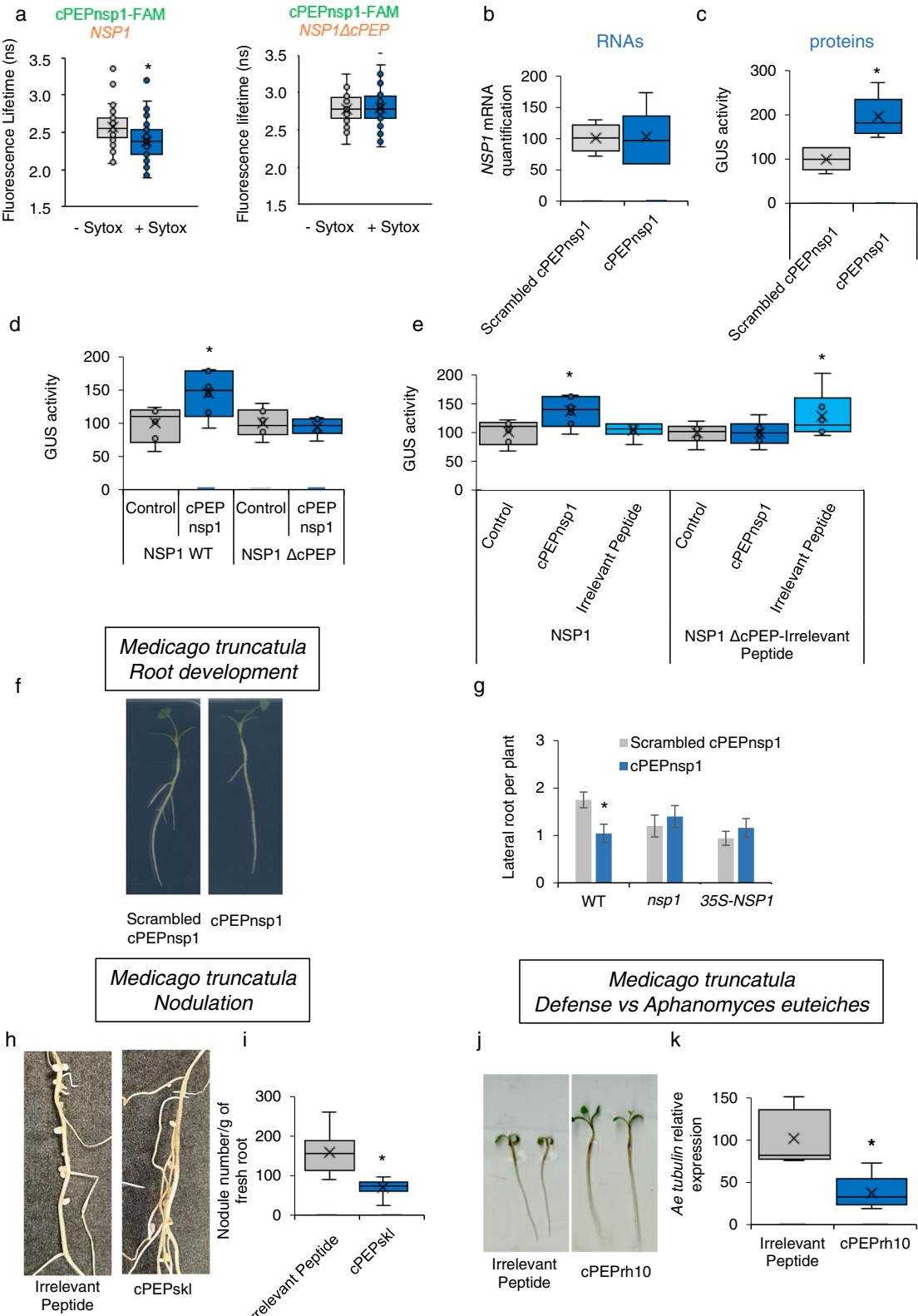

scrambled peptide (Scrambled cPEPcpk3-HA), validated its activity on CPK3 expression (Supplementary Fig. 4a), and performed a co-immunoprecipitation with HA antibodies followed by quantitative MS analysis (Supplementary Fig. 4b). This approach allowed us to identify two proteins, the beta-amylase BAM3 (At4g17090) and the ribosomal protein RPL19 (At4g17560), enriched in eluates of cPEPcpk3-HA immunoprecipitated protein complexes compared to immuno-precipitation with a scrambled peptide. While the role of BAM3 is

currently not understood, RPL19, as a component of the ribosome machinery, is suspected to increase the stability of inter-ribosomal subunit bridges[13] and might participate in cPEP activity. To test RPL19 relevance in vivo, an *rpl19* mutant was treated with either cPEPcpk3 or cPEPdcl1 and the phenotypes were analyzed. Interestingly, the mutant was insensitive to the different peptides (Fig. 4d, e), showing that RPL19 is required for cPEP functions in vivo. Finally, we performed polysome purification from plants treated with cPEPluc-HA or the

**Fig. 2 | cPEPs require sequence complementarity with their target for their activity. a** FRET-FLIM analysis of *in planta* interaction between cPEPnsp1-FAM and *NSP1* (left) or *NSP1ΔcPEP*, an *NSP1* version in which cPEP-corresponding sequence was removed (right) *in planta* (see Suppl. Table 2). **b** Quantification by qRT-PCR of *NSP1* expression in *M. truncatula* roots treated with cPEPnsp1 or Scrambled cPEPnsp1. **c** GUS quantification in *M. truncatula* roots expressing *ProNSP1-NSP1-GUS* fusion and treated with the indicated peptides. **d, e** Quantification of *NSP1* expression in *N. benthamiana* leaves after infiltration of different constructs: wild type or a *NSP1* version in which the cPEP-corresponding sequence was removed (*NSP1 ΔcPEP*) or *NSP1* in which cPEP sequence was replaced by a random artificial sequence (*NSP1 ΔcPEP-Irrelevant cPEP*), together with an empty vector (control) or a vector expressing either cPEPnsp1 or the Irrelevant cPEP. **f, g** Lateral root formation of WT, *nsp1* or overexpressing NSP1, *M. truncatula* seedlings in response to cPEPnsp1 or Scrambled cPEPnsp1. **h, i** Quantification of nodules on *M. truncatula* roots treated with an irrelevant peptide or cPEPskl. **j** Root development of *M. truncatula* seedlings infected by *A. euteiches* and treated with an irrelevant peptide or cPEPrh10. **k** Relative *A. euteiches* α-tubulin expression in *M. truncatula* seedlings infected with *A. euteiches* and treated with an irrelevant peptide or cPEPRh10. For box plots, the cross represents the mean, the line shows the median value, and the upper part and lower part of the box represent the first and third quartiles. The error bars represent the minimal and maximal value for box plots and the standard error of the mean (SEM) for others. Asterisks indicate a significant difference between the test condition and the control according to the Student *t*-test (**a**, **f–i**) or Wilcoxon test (**b–e**, **k**) (**a**, $n = 39$–$62$, **b–e**, **k** $n = 6$; **f–i** $n = 50$; $p < 0.05$). Source data are provided as a Source Data file.

corresponding scrambled peptide. Western blot analysis using HA antibodies revealed that the cPEP (but not the scrambled peptide) was present in the polysomal fractions (fractions 8 to 12), suggesting that cPEPs might interact with ribosomes in general (Fig. 4f).

## cPEPs can modulate traits with agronomical interest

In order to investigate whether cPEPs could be used for agronomic purposes to improve crop yields, we tested them on plants of agronomic interest, focusing on the same phenotypes that we studied in model plants. Thus, we first studied plant defense in tomato by targeting the JAR1 protein. Consistent with the previous observation in *A. thaliana*, treatment of tomato plants with cPEPjar1 was able to improve plant resistance to *B. cinerea* (Fig. 5a, b). In parallel, we identified the homolog of *A. thaliana* HSP101 in soybean and designed a cPEP targeting this protein. Interestingly, treating soybean plants with this peptide increased their tolerance to heat stress (Fig. 5c, d). In addition, we validated on soybean that the use of cPEPs can improve plant growth, using a mixture of cPEPs targeting SHY2, MRB1, and SGR1 (Fig. 5e, f). Finally, we tested whether cPEPs could decrease weed growth by targeting a Brassicaceae weed species, *B. vulgaris*, and showed that a mixture of cPEPs targeting EIN2, BRI1, BAK1, and WUS, was able to do so (Fig. 5g, h). To take this a step further, we chose one of the most invasive and problematic weeds, Amaranthus, and designed cPEPs to target its corresponding proteins. A mixture of these cPEPs was able to decrease plant growth (Fig. 5i, j).

## Discussion

Our findings unveil the possibility to modulate protein expression by external application of small synthetic peptides, facilitating the study of genes without the need for transgenic plants. This may be particularly relevant in the case of plants recalcitrant to genetic transformation. Simply watering or spraying plants with cPEPs allows a biological response consistent with what is known about the function of the targeted proteins, such as modulating plant growth or enhancing plant tolerance to certain pathogens.

The design of cPEPs seems to follow no particular rule, indeed, we designed many peptides targeting luciferase, and all of them were active in increasing luciferase activity. We designed them without any a priori knowledge, except the fact that the peptides must be at least a little hydrophilic, to facilitate their solubilization (see methods). In parallel, all the peptides we tested for their molecular activity led to increased expression of their targeted protein, confirming that there is no rule for designing cPEPs.

Here we described the use of cPEPs to externally modulate the translation of transcripts coding for proteins (Fig. 6). All tested peptides were designed artificially, with the use of bioinformatics, and we cannot exclude that cPEPs exist in *planta*. Whether such peptides hidden in the plant genomes exist still remains to be determined and constitute the next line of research. Several recent findings from other groups demonstrate that many short open reading frames hidden in intergenic sequences or within coding sequences have the ability to produce small functional peptides, possibly natural cPEPs (natcPEPs), both in plants and other species[14–16]. Whether these natcPEPs impinge on the expression of their target protein remains to be determined.

Several natural sources of natcPEPs can be considered. First, long non-coding RNA (lncRNAs), which have already been shown to encode small peptides[14, 15]. A second source of natcPEPs would be the coding genes themselves. Indeed, 5′ and 3′ untranslated regions (UTRs), as well as alternative ORFs (named altprot) present in the main ORFs of coding genes, can encode small peptides[14–16]. Finally, another potential source of natcPEPs might be peptides resulting from protein degradation, produced by the proteasome or other intracellular proteases. An enticing hypothesis would be that short peptides produced by proteases could act as cPEPs in order to compensate for degradation to maintain a steady state level of cPEP-targeted proteins. The main issue in identifying natcPEP sources comes from the difficulty to detect these peptides *in planta*. Indeed, the identification of such small molecules by mass spectrometry still remains challenging[17].

The 21st-century agriculture faces several huge challenges to feed the growing world population. In this context, finding new molecules to keep or improve crop yields is of urgent need. Up to now, no credible alternative to chemicals has emerged. The use of CRISPR in agriculture is promising, especially to improve crop growth, but it is difficult to be applied in weed control. The use of small RNAs, despite its fantastic potential[18], faces the unsolvable problem of poor penetration into plant cells, leading to poor activity in fields, except for insect control[19]. We show here that cPEPs are able to modulate different plant traits, such as heat resistance and chlorophyll content, traditionally difficult to manage with chemicals or other molecules. In this context, the discovery of cPEPs opens a new way in agriculture with the use of small peptides. More importantly, cPEP activity, based on sequence identity with targeted protein, will allow easier bioinformatic identification of off-targets, allowing the target of only one plant species, a plant family, or all plants.

Finally, because peptides are short polymers of amino acids, they are likely to be rapidly degraded by soil microbiota, unlike polluting chemicals. Moreover, the penetration of peptides into animal cells seems difficult without the presence of cell-penetrating peptides[20, 21], suggesting that cPEPs will probably have no biological activity in animals (like bees and butterflies) or humans.

## Methods

### Biological material and growth conditions

*Medicago truncatula* Gaertn cv. Jemalong genotype A17 plants were cultivated on Long Aston medium[22]. *Arabidopsis thaliana* Col-0 plants were grown on Jiffy up to 4 weeks old in a growth chamber (22/20 °C, 16 h/8 h light/dark, RH 80%, ~75 µmol m$^{-2}$ s$^{-1}$). ABRE-LUC seedlings were provided by MR Knight[6]. Concerning in vitro experiments, surface-sterilized Col-0 seeds were sown onto ½ MS solid medium and stratified for 24 h at 4 °C in the dark (same conditions as above). Seedlings were vertically grown in a controlled growth chamber. *Barbarea vulgaris* seeds were stratified for 24 h at

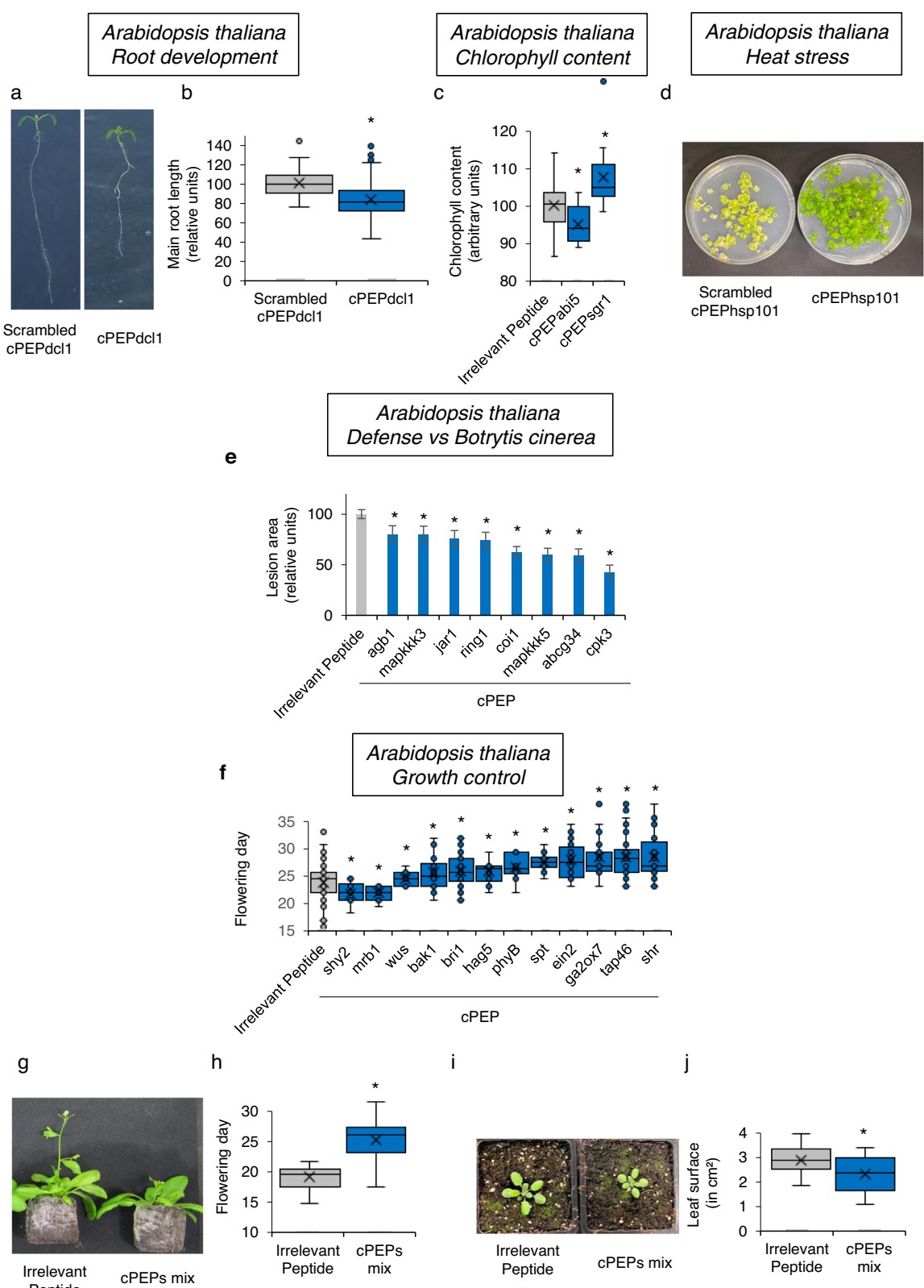

4 °C before being cultivated on a pot in a growth chamber (same conditions as above). *Amaranthus hypochondriacus, Glycine max, N. benthamiana*, and *Solanum lycopersicum* seeds were sowed on pots and cultivated in growth chamber. *Medicago truncatula* seeds were sterilized with bleach and sowed on pots, and cultivated in a growth chamber[23]. Inoculation with *Sinorhizobium meliloti* 2011 was performed one week after germination with 10 mL of a bacteria suspension at OD = 0.05. *Arabidopsis thaliana* LUC plants were sterilized and sowed on 96 wells plates containing 100 μL of MS/2 medium.

**Peptides**

Peptides were synthesized by Smart Biosciences (www.smart-bioscience.com) and dissolved at 2–10 mM in water, aliquoted and

**Fig. 3 | cPEPs can modulate *A. thaliana* development and response to stresses.** **a**, **b** Primary root length of *A. thaliana* seedlings treated with Scrambled cPEPdcl1 or cPEPdcl1. **c** Relative chlorophyll content of *A. thaliana* plants treated with the indicated peptide. **d** Growth recovery of *A. thaliana* seedlings after a heat shock of 45 °C for 45 min and treated with the indicated peptide. **e** Relative lesion area of *A. thaliana* plants infected by *B. cinerea* and treated with the indicated peptide. **f** Measurement of the flowering day of *A. thaliana* plants treated with the indicated peptides. **g**, **h** Measurement of the flowering day of *A. thaliana* plants treated with an irrelevant peptide or a mix of cPEPs (targeting EIN2, BRI1, BAK1, and WUS).

**i**, **j** Leaf surface of *A. thaliana* plants treated with an irrelevant peptide or a mix of cPEPs (targeting EIN2, BRI1, BAK1, and WUS). For box plots, the cross represents the mean, the line shows the median value, and the upper part and lower part of the box represent the first and third quartiles. The error bars represent the minimal and maximal value for box plots and the standard error of the mean (SEM) for others. Asterisks indicate a significant difference between the test condition and the control according to the Student *t*-test (**b**, **c**, **e**, **h**, **j**) (**b** $n = 200$; **c**, **e**, **f**, **h**, **j** $n = 40$; **d** $n = 5$; $p < 0.05$). Source data are provided as a Source Data file.

conserved at −80 °C. All the peptides are listed in Supplementary Data 1. All cPEPs were designed to be hydrophilic, i.e., contain at least more than 33% of the following amino acid: D, N, E, Q, K, and R.

### Expression analyses
Gene expression quantification was performed by qRT-PCR. Levels of expression for the controls were set at 100. Primers used for *A. euteiches* assays were described in ref. [9]. List of primers used in this study was listed in Supplementary Table 3.

### Pathogenicity assays
Wild-type or mutant *A. thaliana* leaves were sprayed daily with 100 μM of peptide or 100 μM related scramble for 3 days. Six hours after the last treatment, five mature leaves per plant were inoculated with a 5 μL droplet of $2.5 \times 10^5$ spores/mL of *Botrytis cinerea* strain B05.10 diluted into 100 μM of peptide or its scrambled version. After inoculation, plants were kept at 100% relative humidity. Then, a 2 μL droplet of 100 μM peptide or 100 μM related scramble was put daily onto *B. cinerea* infected leaves for 3 days (until symptom appearance). Leaves were excised from plants to determine lesion areas by using the ImageJ program. For tomatoes, the protocol was the same, except that inoculation was performed with 5000 spores of *B. cinerea*, without peptide. The peptide was added 1 h after inoculation (1 μL of 500 μM) and every day for two days with 5 μL of 100 μM peptide.

For *M. truncatula* infection with *A. euteiches*, plants were cultivated on agar medium and treated with 10 μL of 100 μM peptide, 24 h before, 24 h and 72 h after infection with 10 μL (1000 spores) of *A. euteiches* spores. Plants were harvested 7 days after inoculation for RNA extraction.

### cPEP treatments
*N. benthamiana* plants were treated by spraying leaves 24 h before harvesting. For Luc assays, 100 μL of MS/2 liquid medium containing peptide was added to each well. Five microliters of luciferin was added and luciferase activity was read 30 min later using a spectrophotometer. All the other assays were performed by spraying or watering plants. For quantification of cPEP-induced protein level in *Arabidopsis*, rosettes were sprayed every day with 100 μM of peptide or its control for 5 days. Leaves were harvested 6 h after the last treatment for western blot analyses or mass spectrometry analyses. Concerning flowering assay and chlorophyll content, 10-day-old *Arabidopsis* plants were sprayed with 500 μL of 10 μM of peptide, three times a week until bolting. Chlorophyll content was measured with a SPAD chlorophyll meter (Konica Minolta). The leaf surface was measured using ImageJ software.

Concerning Arabidopsis root development, Col-0 seeds were grown in vitro as described above. Three days after sowing, seedlings were treated with 100 μM of the corresponding cPEP or its scramble every 2 days for 2 weeks. Seedlings were harvested 24 h after the last treatment and scanned to quantify the primary root length using the NeuronJ plugin of ImageJ.

For growth experiments, *B. vulgaris, A. hypochondriacus, A. thaliana*, and *G. max* seedlings were treated just after sowing and three times a week with 500 μL of a mix of 20 μM of each peptide. For western blot, heat shock was performed by placing 20 days old plants

grown in 24-well plates on MS/2, 90 min at 37 °C before harvesting. Plants were treated with 100 μM peptide for 24 h. For heat shock resistance assay on *A. thaliana*, 3 days old seedlings were treated for 3 days with 100 μM peptide before placing plants at 45 °C for 45 min. Plants were put in a growth chamber for recovery and treated for 24 h after heat shock with 100 μM peptide. For soybean heat shock, 1-week-old seedlings were treated 48 h, 24 h, and 30 min before placing them at 45 °C for 24 h. For soybean growth, 1-week-old plants were treated three times a week for 2 weeks with 500 μL of 100 μM of peptides.

### Protein extraction and western blot analyses
Total proteins from *Arabidopsis* plants were extracted by centrifugation and resuspension in buffer, before Western Blot[24], with the following antibodies: anti-HA (Sigma Aldrich; H6908; 1:10,000), anti-GUS (Abcam; ab50148; 1:5,000), anti-LUC (Agrisera; AS16 3691 A; 1:1,000), anti-Ein2 (Agrisera; AS12 1865; 1:10,000; also recognize SKL from *M. truncatula*), anti-Bak1 (Agrisera; AS12 1858; 1:2,000), anti-Bri1 (Agrisera; AS12 1859; 1:5,000), anti-DCL1 (Agrisera; AS19 4307; 1:1,000), anti-HSP101 (Agrisera; AS07 253; 1:1,000) anti-GFP (Sigma Aldrich; SAB4301138; 1:5,000). Anti-CPK3 and anti-GAPC antibodies were raised in rabbits by Covalab (France) using purified recombinant CPK3-GST and purified recombinant *At*GAPC1-GST, respectively. Anti-CPK3 antibodies were purified from rabbit serum by affinity chromatography on 6His-CPK3 coupled to CH-sepharose (GE Healthcare), while crude serum was used for anti-GAPC western blots. Anti-CPK3 and anti-GAPC were used at a dilution of 1:5000 and 1:50,000, respectively. Protein levels were normalized to Ponceau staining.

### In vitro transcription/translation
The TnT® SP6 High-Yield Wheat Germ Protein Expression System (Promega) was used according to the instructions of the manufacturer. LUC sequence was amplified using the Gotaq polymerase (Promega), and the cPEP or related scrambled version were added just before the beginning of the reaction. The reaction was stopped after 60 min, after which the LUC activity was measured with a plate spectrophotometer (Perkin-Elmer Victor Nivo).

### Cycloheximide treatment
*Arabidopsis thaliana* plants carrying a LUC construct were treated with peptide solution with or without cycloheximide (200 μg mL$^{-1}$) in MS/2 medium, for 24 h before harvesting for LUC assay.

### RNA IP
RNA co-immunoprecipitation was adapted from ref. [25]. About 400 mg of tissue powder were incubated in 3 mL of lysis buffer (200 mM Tris, pH 9.0, 110 mM potassium acetate, 0.5% Triton X-100, 0.1% Tween 20, 5 mM DTT, 1.5% protease inhibitor and 80 units ml$^{-1}$ RNasin). The lysate was incubated on ice for 10 min and then centrifuged at $16,000 \times g$ for 10 min at 4 °C. About 1.5 mL of crude extract was incubated with 25 μL of anti-HA magnetic beads (Thermo Scientific) for 1.5 h at 4 °C under rotation. After binding, beads were washed five times with 0.75 mL of lysis buffer. Elution was performed with 200 μL of 8 M guanidium for 5 min on ice and precipitated overnight with 300 μL of 100% ethanol. After

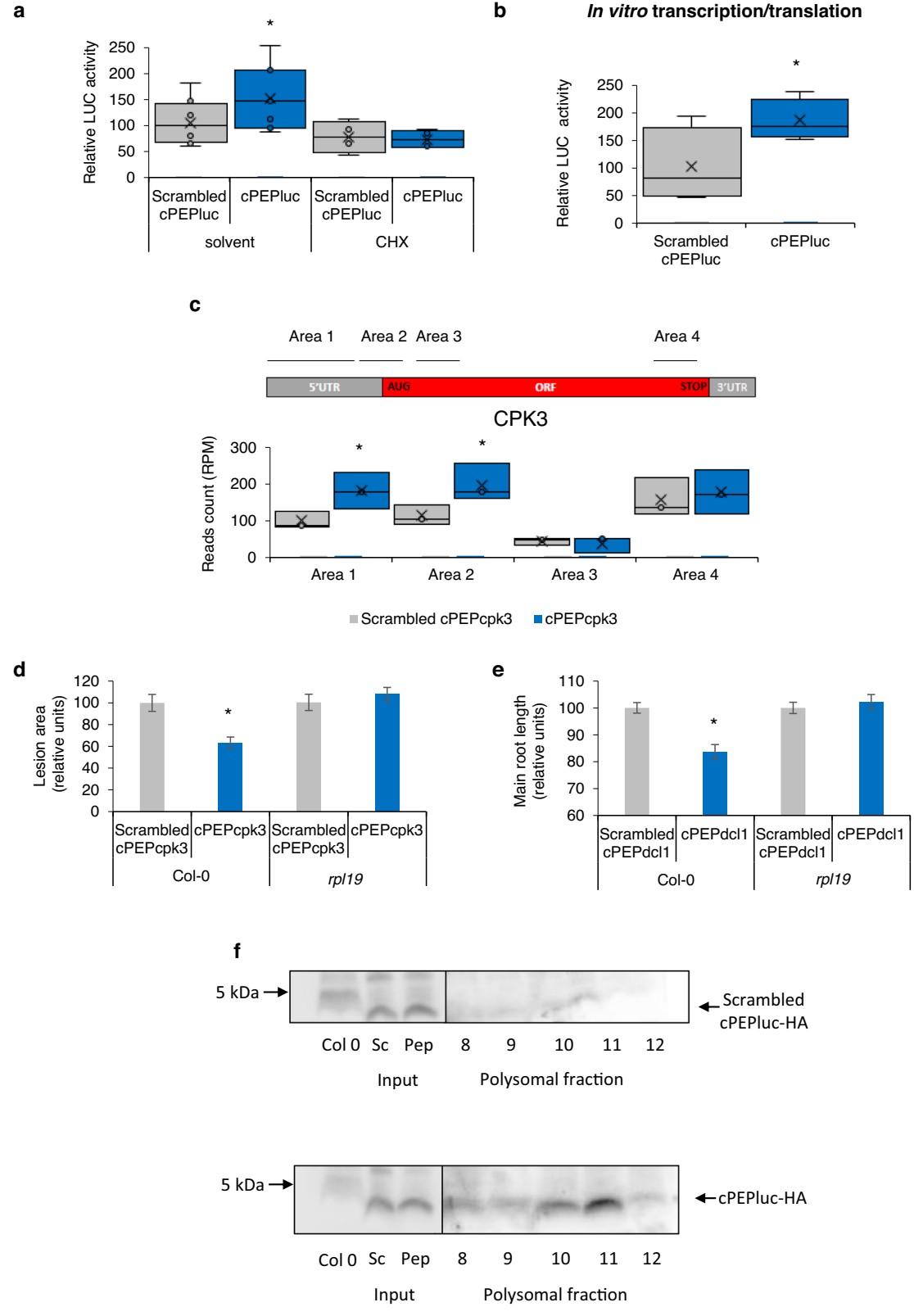

**Nature Communications** | (2023)14:254

centrifugation (16,000×*g*, 45 min, 4 °C), pellets were resuspended in 200 µL of Monarch DNA/RNA Protection Reagent (New England Biolabs) and RNA was extracted according to manufacturer's instructions and concentrated in 10 µL using Monarch RNA Cleanup Kit (New England Biolabs). A total of 300 µL of input and unbound fractions were kept and RNA was extracted as described above. Reverse-transcription was performed on 10 µL of eluate or 500 ng of input/unbound using a Superscript IV kit (Thermo Scientific). PCR amplification was performed on 1 µL of cDNA with specific primers.

**FRET-FLIM**

Samples were prepared based on the following protocol[26]: *Agrobacterium tumefaciens* GV3101pmp90 strain carrying *35S-NSP1* or *35S-NSP1ΔORF* (*35S-NSP1ΔcPEP*) plasmids were used to infiltrate *N.*

**Fig. 4 | cPEPs increase protein translation. a** Analysis of luciferase activity in *A. thaliana* plants constitutively expressing *LUC* transgene and treated with a Scrambled cPEPluc or cPEPluc, with or without cycloheximide. **b** Analysis of luciferase activity after in vitro transcription/translation of luciferase expressed together with a Scrambled cPEPluc or cPEPluc. **c** Quantification of 5′P reads accumulation in cPEPcpk3-HA treated *A. thaliana* plants on four distinct areas along transcripts, expressed as CPK3 read counts (RPM value) in each area. Area 1: −60 nt to −15 nt, Area 2: −14 nt to 31 nt, Area 3: 32 nt to 77 nt from the start codon, Area 4: −62 nt to −17 nt from the stop codon. **d** Infection assay of *A. thaliana* Col-0 and *rpl19* mutant leaves inoculated with *B. cinerea* spores and treated with a Scrambled cPEPcpk3 or cPEPcpk3. **e** Primary root length of *A. thaliana* Col-0 and

*rpl19* mutant seedlings treated with a Scrambled cPEPdcl1 or cPEPdcl1. **f** Western blot analysis using HA antibody of polysomal fractions (8–12) extracted from *A. thaliana* plants constitutively expressing *LUC* transgene and treated with a Scrambled cPEPluc-HA or cPEPluc-HA. For box plots, the cross represents the mean, the line shows the median value, and the upper part and lower part of the box represent the first and third quartiles. The error bars represent the minimal and maximal value for box plots and the standard error of the mean (SEM) for others. Asterisks indicate a significant difference between the test condition and the control according to the Student *t*-test (**d**, **e**) or to the Wilcoxon test (**a**–**c**) (**a**, **b** *n* = 6; **c**, **f** *n* = 3; **d** *n* = 20, **e** *n* = 200; *p* < 0.05). Source data are provided as a Source Data file.

*benthamiana* leaves. After 40 h, a 10 μM cPEP-FAM solution was infiltrated in the same leaves and plants were incubated for 3 h. Then infiltrated disks from at least three different leaves were fixed by vacuum-infiltrating a 4% (w/v) paraformaldehyde solution, followed by a permeabilization step using a proteinase K treatment. After washes with TBS, nucleic acid staining was performed by vacuum-infiltrating a 5 μM Sytox Orange (Invitrogen) solution. Then disks were washed and mounted on TBS before cytoplasmic −FRET-FLIM measurements.

FLIM was performed on Leica TCS SP8 SMD, which consists of an inverted LEICA DMi8 microscope equipped with a TCSPC system from PicoQuant. The excitation of the FITC donor at 470 nm was carried out by a picosecond pulsed diode laser at a repetition rate of 40 MHz, through an oil immersion objective (63×, N.A. 1.4). The emitted light was detected by a Leica HyD detector in the 500−550 nm emission range. Images were acquired with acquisition photons of up to 1500 per pixel.

From the fluorescence intensity images, the decay curves were calculated per pixel and fitted (by Poissonian maximum likelihood estimation) with either a mono- or double-exponential decay model using the SymphoTime 64 software (PicoQuant, Germany). The mono-exponential model function was applied for donor samples with only FITC present. The double-exponential model function was used for samples containing FITC and Sytox. Experiments were repeated at least three times to get statistically valid data. The efficiency of energy transfer (*E*) based on the fluorescence lifetime (τ) was calculated as

$$E = 1 - (\tau_{D+A}/\tau_{D-A}) \qquad (1)$$

where $\tau_{D+A}$ is the donor fluorescence lifetime in the presence of the acceptor while $\tau_{D-A}$ is the donor fluorescence lifetime in the absence of the acceptor.

## IP-mass spectrometry

To immunoprecipitate proteins associated with cPEPcpk3-HA or its corresponding HA-tagged scrambled peptide, 3 mg of *Arabidopsis* total proteins were incubated with 40 μL μMACS anti-HA microbeads (Miltenyi Biotec) for 1 h. After this incubation, washing was performed using the manufacturer's instruction (Miltenyi Biotec). Beads were resuspended in 150 μL of Tris HCl (pH 7.5) 20 mM for further analyses.

Washed and resuspended beads were incubated for 4 h with 1 μg trypsin (Promega) at 37 ˚C. Beads were removed, another 1 μg of trypsin was added and proteins were further digested overnight at 37 ˚C. Peptides were purified on Omix C18 tips (Agilent), dried, and re-dissolved in 20 μL loading solvent (0.1% trifluoroacetic acid in water/acetonitrile (ACN) (98:2, v/v) of which 6 μL were injected for LC-MS/MS analysis on an Ultimate 3000 RSLCnano system (Thermo) in-line connected to a Q Exactive mass spectrometer (Thermo). Trapping was performed at 10 μL/min for 4 min in loading solvent A on a 10 mm μPAC™ trapping column (PharmaFluidics) with C18-endcapped stationary phase and the samples were loaded on a 50 cm long micropillar array column (PharmaFluidics) with C18-endcapped functionality mounted in the Ultimate 3000's column oven set at 35 °C. For proper ionization, a fused silica PicoTip emitter (10 μm inner diameter, New Objective) was connected to the μPAC™ outlet union and a grounded

connection was provided to this union. Peptides were eluted by a non-linear increase from 1 to 50% MS solvent B (0.1% formic acid (FA) in water/ACN (2:8, v/v)) over 99 min, first at a flow rate of 750 nl/min, then at 300 nl/min, followed by a 5-min wash reaching 99% MS solvent B and re-equilibration with MS solvent A (0.1% FA in water). The mass spectrometer was operated in data-dependent, positive ionization mode, automatically switching between MS and MS/MS acquisition for the 5 most abundant peaks in a given MS spectrum. The source voltage was 2.7 kV, and the capillary temperature was 275 °C. One MS1 scan (*m/z* 400 − 2000, AGC target $3 \times 10^6$ ions, maximum ion injection time 80 ms), acquired at a resolution of 70,000 (at 200 *m/z*), was followed by up to five tandem MS scans (resolution 17,500 at 200 *m/z*) of the most intense ions fulfilling predefined selection criteria (AGC target $5 \times 10^4$ ions, maximum ion injection time 80 ms, isolation window 2 Da, fixed first mass 140 *m/z*, spectrum data type: centroid, underfill ratio 2%, intensity threshold $1.3 \times E^4$, exclusion of unassigned, 1, 5-8, >8 positively charged precursors, peptide match preferred, exclude isotopes on, dynamic exclusion time 12 s). The HCD collision energy was set to 25% Normalized Collision Energy and the poly-dimethylcyclosiloxane background ion at 445.120025 Da was used for internal calibration (lock mass).

Data analysis was performed with MaxQuant (version 1.6.3.4) using the Andromeda search engine with default search settings, including a false-discovery rate set at 1% on both the peptide and protein levels. Spectra were searched against the annotated sequences of the *A. thaliana* Col-0 genome (Araport11_TAIR11 containing 48,359 protein sequences), supplemented with peptide sequences (cPEPcpk3-HA and scrambled cPEPcpk3-HA). The mass tolerance for precursor and fragment ions was set to 4.5 and 20 ppm, respectively, during the main search. Enzyme specificity was set as C-terminal to arginine and lysine, also allowing cleavage at proline bonds with a maximum of two missed cleavages. Variable modifications were set to oxidation of methionine residues and acetylation of protein N-termini. Matching between runs was enabled with a matching time window of 1.5 min and an alignment time window of 20 min. Only proteins with at least one unique or razor peptide identified were retained, leading to the identification of 712 proteins. Proteins were quantified by the MaxLFQ algorithm integrated into the MaxQuant software. A minimum ratio count of two unique or razor peptides was required for quantification. Further data analysis was performed with the Perseus software (version 1.6.2.1) after loading the protein groups file from MaxQuant. Reverse database hits were removed and replicate samples were grouped. Proteins with less than three valid values in at least one group were removed and missing values were imputed from a normal distribution around the detection limit leading to a list of 463 quantified proteins that was used for further data analysis. On these quantified proteins, a *t*-test was performed for a pairwise comparison of both conditions. The results of this *t*-test are shown in the volcano plot (Supplementary Fig. 4b). For each protein, the log₂ (HA-tagged peptide/HA-tagged scrambled peptide) fold change value is indicated on the X-axis, while the statistical significance (−log *p* value) is indicated on the Y-axis. Proteins outside the curved lines, set by an FDR value of 0.05 and an S0 value of 1 in the Perseus software, represent specific

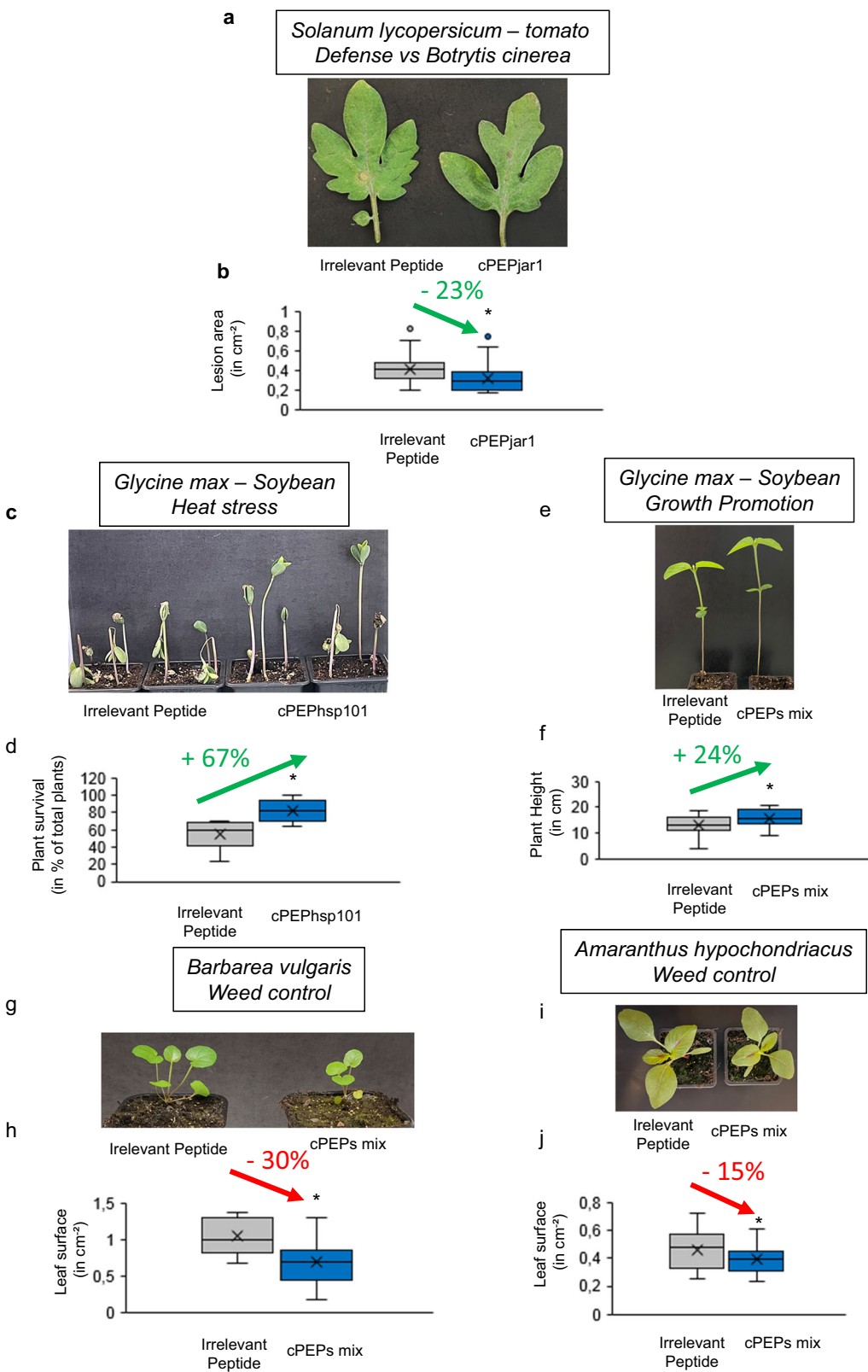

**Fig. 5 | cPEPs are useful tools in agronomy. a**, **b** Relative lesion area of *S. lycopersicum* leaves infected with *B. cinerea* and treated with an irrelevant peptide or cPEPjar1. **c**, **d** Resistance to heat stress of soybean plants treated with an irrelevant peptide or cPEPhsp101. **e**, **f** Growth (plant height) of soybean plants treated with an irrelevant peptide or a mixture of cPEPs (targeting MRB1, SHY2, and SGR1). **g**, **h** Leaf surface of *B. vulgaris* plants treated with an irrelevant peptide or a mixture of cPEPs (targeting EIN2, BRI1, BAK1, and WUS). **i, j** Leaf surface, of *A. hypochondriacus* plants treated with an irrelevant peptide or a mixture of cPEPs (targeting EIN2, BRI1, BAK1,

and WUS). For box plots, the cross represents the mean, the line shows the median value, and the upper part and lower part of the box represent the first and third quartile. The error bars represent the minimal and maximal value for box plots and the standard error of the mean (SEM) for others. Asterisks indicate a significant difference between the test condition and the control according to the Student *t*-test (**b**, **d**, **f**, **h**, **j** $n = 40$; $p < 0.05$). Source data are provided as a Source Data file.

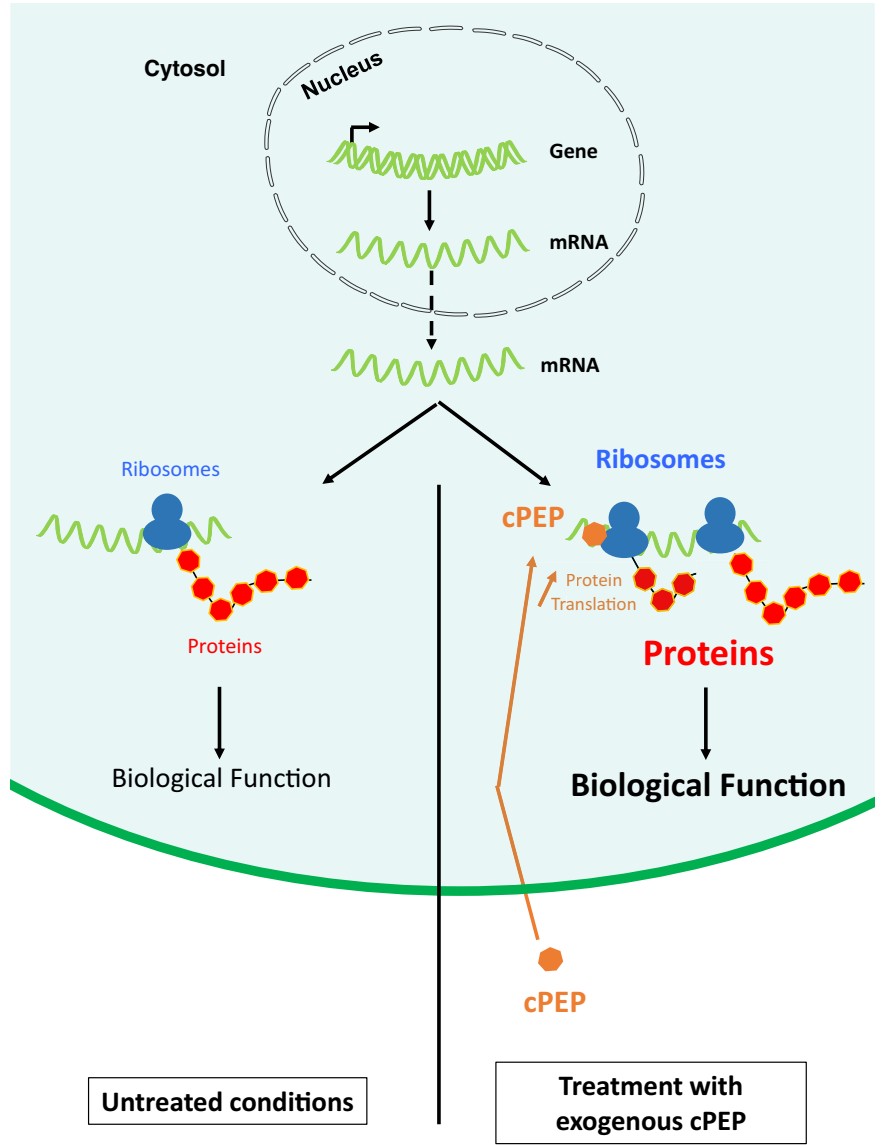

**Fig. 6 | Schematic representation of the mode of action of cPEPs.** cPEPs exogenously applied on plants penetrate plant cells and interact with their own RNA and ribosomes to improve their recruitment and increase the translation of targeted proteins. This leads to phenotypes consistent with the known role of targeted proteins.

HA-tagged peptide interaction partners. The mass spectrometry proteomics data have been deposited to the ProteomeXchange Consortium via the PRIDE partner repository with the dataset identifier PXD017404 (ProteomeXchange Dataset PXD017404).

### 5′P sequencing
Extraction was performed on powder using Monarch Total RNA Miniprep Kit (New England Biolabs) using the manufacturer's instructions. 5′P libraries were prepared and analyzed through GMUCT assay[12] with minor modifications. Briefly, 50 µg of total RNA was used as a starting point. After PolyA+ purification, mRNAs were subjected to 5′ adapter ligation and reverse-transcription using Illumina Small RNA library Kit. Libraries were amplified using 11 cycles of PCR and purified using SPRI beads. Libraries were multiplexed and sequenced (SR75) on an Illumina NextSeq 550. For each library, at least 60 million reads were obtained.

Prior to analysis, reads were trimmed to 50 bp using Trimmomatic[27]. Reads were then aligned against TAIR10 genome annotation using HiSat2[28]. Subsequently, the BAM files were converted to BED files containing only the first nucleotide of each read. Reads count was performed on a defined area and normalized according to sequencing depth with BEDtools suite[29].

### Polysome analysis
Polysomes analysis was done with three biological replicates for each control and treatment. Polysomes were extracted and separated on a sucrose gradient[25]. Each gradient was fractionated into ten fractions, and the proteins were extracted and analysed by SDS-page gel blotting.

### Proteomics
Four weeks old luciferase expressing plants[6] were sprayed with 50 µM cPEPluc-HA or Scrambled cPEPluc-HA. Luciferase assay was performed

to check cPEP effect 24 h later. Rosettes were then harvested, frozen in liquid nitrogen, and grounded into powder.

The powder was resuspended in Tris 50 mM and 5% SDS, sonicated, and centrifuged at 14,000×g for 10 min. A BCA protein assay was performed on the supernatant to estimate protein concentration in each sample. About 50 μg of proteins were then loaded on a 12% Tris-glycine gel. Proteins were concentrated into one band and gel digestion was performed[30]. The peptides are resuspended in 14 μL of 2% Acetonitrile, 0.05% TFA, vortexed, and sonicated for 10 min before injection. About 500 ng of the peptides mixtures were analysed by nano-LC-MS/MS using nanoRS UHPLC system coupled to an Orbitrap Exploris 480 mass spectrometer using FAIMS Pro Duo interface (Thermo Fisher Scientific, Bremen, Germany). One microliter of each sample were loaded on an analytical C18 column (PEPMAP C18 2 μm 75 μm × 500 mm Thermo Fisher Electron) equilibrated in 95% of solvent A (5% acetonitrile + 0.2% formic acid in water) and 5% of solvent B (80% acetonitrile + 0.2% formic acid in water). Peptides were eluted using a 5–50% gradient of B for 130 min at a 300 nL/min flow rate. The Orbitrap Exploris 480 was operated in FAIMS mode (two compensation voltages used: −45 and −60 v) and data-dependent acquisition mode with the Xcalibur software. Survey scan MS spectra were acquired in the Orbitrap on the 375–1200 $m/z$ range with the resolution set to a value of 60,000. The ten most intense ions per survey scan were selected for HCD fragmentation, and the resulting fragments were analysed in the Orbitrap with the resolution set to a value of 15,000. Dynamic exclusion was used within 45 s to prevent repetitive selection of the same peptide.

Acquired MS and MS/MS data were searched with Mascot (version 2.8.0, http://matrixscience.com) against a custom-made database containing all Arabidopsis entries from the UniprotKB database, several interest protein sequences, and contaminants protein sequences. The search included methionine oxidation, and acetylation N terminal, as variable modifications, and carbamidomethylation of cysteine as a fixed modification. Trypsin P was chosen as the enzyme and two missed cleavages were allowed. The mass tolerance was set to 20 ppm for the precursor ion and 20 mmu for fragment ions. Raw MS signal extraction of identified peptides was performed across different samples. Validation of identifications was performed through a false-discovery rate set to 1% at protein and peptide-sequence match level, determined by target-decoy search using the in-house-developed Proline software version 2.1 (http://proline.profiproteomics.fr/)[31].

### Statistical analyses

The mean values of relative gene expression, protein level, or phenotypical parameters were compared by using the Wilcoxon or the Student $t$-test. For box plots, the cross represents the mean, the line shows the median value, and the upper part and lower part of the box represent the first and third quartiles. The error bars represent the minimal and maximal value for box plots and the standard error of the mean (SEM) for others. The asterisks indicate significant differences between the test and the control ($p < 0.05$). In all the figures, '$n$' corresponds to biological replicates.

### Reporting summary

Further information on research design is available in the Nature Portfolio Reporting Summary linked to this article.

## Data availability

The 5 P sequencing data generated in this study have been deposited in the NCBI database under accession PRJNA605186. The MS data generated in this study have been deposited in the MassIVE database under accession MSV000090305 [https://massive.ucsd.edu/ProteoSAFe/dataset.jsp?task=a7eb661040c74675abf409a0fbfc240a]. The IP MS data generated in this study have been deposited in the ProteomeXchange Consortium database under accession code PXD017404. Source data are provided with this paper.

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

## Acknowledgements

This work was funded by the French ANR project BiomiPEP (ANR-16-CE12-0018-01). B.F. has been supported by the Fondation ARC pour la recherche sur le cancer. The work (proteomics) was funded in part by grants from the Région Occitanie, European funds (Fonds Européens de Développement Régional, FEDER), Toulouse Métropole, and the French Ministry of Research with the Investissement d'Avenir Infrastructures Nationales en Biologie et Santé program (ProFI, Proteomics French Infrastructure project, ANR-10-INBS-08). We thank Christian Mazars (LRSV, Auzeville-Tolosane, France) for valuable advice about CPK3, Marc Knight (Durham University, UK) for Arabidopsis ABRE Luciferase lines, and Camille Ribeyre (Micropep Technologies) for *Botrytis cinerea* spores. This work was supported by UPVD University through the utilization of the NextSeq 550 device at the Bioenvironment Platform.

## Author contributions

J.-P.C designed the research; J.-P.C., B.G., D.L., L.C., M.O., P.T., and C.D. performed the molecular biology and plant experiments; L.C. and C.Po. performed FRET-FLIM analysis, R.M., M.-C.C., and C.B.-A. performed GMUCT and RNA IP analysis; F.I., D.V.H., and K.G. performed MS analysis; B.F., C.Pi., and O.B.-S. performed proteomics analysis, L.B.S. performed polysome analysis; H.S.C. and M.A. performed bioinformatics; J.-P.C., P.T., M.O., and S.P. wrote the paper.

## Competing interests

The authors declare no competing interests.
