## [Peer Review File · Nature Communications]

Complementary peptides represent a credible alternative to agrochemicals by activating translation of targeted proteinsReviewers' Comments:

Reviewer #1:

Remarks to the Author:

The manuscript from Ormancey and colleagues entitled "Complementary peptides (cPEPs): a credible alternative to chemicals in agriculture" report a novel tool, "cPEP", and showed its potential in the regulation of corresponding genes and associated functions. To show the role of cPEP, authors have covered crucial plant processes starting from development to stress response. These findings revealed the possibility of modulating the expression of any coding gene by external application of synthetic peptides. The study is novel as the role of small peptides in enhancing the translation of their own ORFs has not been reported as yet. The study will not only interest groups working in the area of plant science but also open a new area on the regulatory aspect of gene expression across biological sciences. The identification and use of key cPEPs can enhance crop yield and tolerance towards various stresses as well as address weed growth management as an alternative approach to pesticides and chemicals.

Though the manuscript content is novel and needs timely publication, authors need to perform additional experiments to provide more evidence to support their conclusions. My comments are as follows.

Major comments

1. What is the criteria for selecting cPEPs for a gene? Though authors have shown one example of Luciferase, but it is not much clear. The authors should discuss and explain it in detail.
2. To strengthen the hypothesis and results, authors should use Arabidopsis mutants of a few genes to show that cPEP treatment does not have any effect on mutants.
3. Authors need to come out with some hypotheses in relation to the action of cPEPs to enhance the translation of mRNAs. Authors should make efforts to prove the proposed hypothesis.
4. The authors have only two figures in the main document. As the study has novelty, authors need to divide figures (at least into 4) with detailed legends so that readers can understand the novelty of the work.
5. On the basis of the work done, authors need to make a model for the hypothesis and use it as a figure.

Minor Comments

1. In the figure 1 legend, the authors have used the term irrelevant peptide and artificial peptide. What do these terms exactly stand for? Please elaborate on it for better understanding.
2. In one experiment, the authors used a mixture of different peptides. Whether the authors used each peptide with the same concentration or the concentration of the mixture was taken into the consideration.
3. In line no. 94, there is a typo error in fig. 4i, j it should be corrected Fig2i, j
4. Add at least one reference in the statement mentioned in lines 27-33.
5. Regarding statistical analysis in fig 2a, the lesion area of cpk3 seems more significant; please re-check it.

Reviewer #2:

Remarks to the Author:

The paper of Ormancey et al., was initiated by the earlier work of these authors concerning influence of miPEPs on the transcription of pri-miRNAs. It has been shown that miPEPs act as transcriptional

activators responding to the cis-elements which represent sequences of the synthesized RNA chains including the own coding ORFs. Undoubtedly, this is pioneering study of great impact. However, the present work makes a twofold impression.

Evidently, biotechnological perspectives of the observed phenomenon could be substantial. On the other hand, fundamental basis of the research is weakly supported. First, the authors show that the effect of synthetic cPEPs targeting luciferase gene in all 3 ORFs and specifically capable to interact with luciferase mRNA is not connected with mRNA abundance (i.e. transcription). This is in contradiction with the published data of these authors obtained on pri-miRNA+miPEP model.

I feel this contradiction requires comprehensive set of experiments to support understanding the potential mechanisms of action of cPEPs. First, the experiments should be done to discriminate between nucleus-localized and cytoplasm-localized mechanisms. Particularly, in nucleus there can be effect on RNA processing (capping and polyadenylation) as well as on mRNA transfer through nuclear envelope to cytoplasm. All the above processes may influence the amount of synthesized enzyme without increasing the total mRNA concentration. In cytoplasm, the exogenic cPEP may bind to its coding mRNA sequence and somehow may expedite translation. Particularly, this binding could influence the structure of the cytoplasmic luciferase-specific non-translatable mRNPs and makes probable dormant mRNPs more accessible for ribosomes.

Anyway, the additional experiments should be done irrespective actual mechanism of the cPEP action, and this paper cannot be accepted in its present form.

Reviewer #3:

Remarks to the Author:

This study firstly reported the cPEP could induce the relative protein abundance. cPEP_{luc} can specifically target LUC proteins and increase the abundance of the targeted protein, and the authors have already bear out the best maximal effect of cPEP treatment included concentration, time and amino acid length. Based on this results, there were more proteins were chosen, and the relative cPEP can also induce target protein abundance. It means that cPEP is broad-spectrum. In order to verify the specific functions of these cPEPs, they proved that the increased protein abundance after cPEP treatment were sufficient to modulate plant development. Such as enhanced plant defense against the necrotrophic pathogen, decreased chlorophyll content, decreased flowering days and leaf growth. All data showing a synergistic effect of cPEPs, and it is significant to regulate several plant phenotypes. I think this finding is interesting and informative for the plant/crops grow regulation. The external application of synthetic peptides cPEP can be used for agronomic purposes to improve crop yield and decrease weed growth. And these types of peptides represent a credible alternative to chemicals. This report gave us a new method to face the biotic and abiotic stresses, it also contributed to the crop breeding and cultivation. While the following suggestions should be considered:

Q1: In this work, the authors listed the predicted small peptides which encoded by 3 ORF. For LUC protein, they selected several parts of the sequence, and all of the sequence works well. But for the growth or stress response genes, only one sequence was selected to carry out the experiment, what is the criteria for selecting the cPEP sequence for the endogenous genes? If the ORF2 encoded small peptides were selected to treat the plant, will it get similar results? Or the results will be same as the LUC protein.

Q2: the cPEP sequences of the 12 endogenous genes should be supplied in the supplementary data.

Q3: Have you considered the poor specificity of 5-10 amino acids among all proteins in plants? Since they may induce other protein abundance. And considering the diversity of protein and amino acid between different species, the cPEP of model plant has the same effect in other plants? As I see, when we did research on circRNA, we overexpressed the linear sequence of circRNA, sometimes it also got a phenotype of induced stress resistance (but not all of the linear sequence can increase stress resistance). Thus, do you think it can be explained by your report? Did you find other endogenous

genes which protein abundance can not be induced by cPEP?

Q4: In Fig.1 b-e, you use "Scrambled Cpep", but in Fig.2 you use "Irrelevant Peptide". So What is the different between "Scrambled" and "Irrelevant"? And there are some writing mistake in the figures for the word "Irrelevant", for example, Fig1c,d.

Q5: It is recommended that pictures in Fig.2 cd, ef, ig and Extended data Fig.2 gh, ig be arranged horizontally in the same order as another picture.

Reviewer #4:

Remarks to the Author:

In this manuscript the authors demonstrate that a peptide with a sequence which is complementary to that of a protein can result in enhanced translation of the target protein. The results are convincing, but a mechanistic model explaining the action of these cPEP is missing. The authors make reference to their previous evidence that miPEP derived from ORFs in miRNA gene can lead to enhanced transcription of the corresponding miRNA, but here the mechanisms appear to be different, being transcription of the cPEP-target gene unaffected. Interaction of the cPEP with the LUC mRNA is shown, but how is this related to enhanced translation? How does the cPEP interact with its corresponding mRNA? All these questions are not even addressed in the manuscripts.

I would suggest the authors to utilize box-plots instead of histograms to represent their data. Why is SEM used instead of SD when doing the statistical analysis?

REVIEWER COMMENTS

Reviewer #1 (Remarks to the Author):

The manuscript from Ormancey and colleagues entitled “Complementary peptides (cPEPs): a credible alternative to chemicals in agriculture” report a novel tool, “cPEP”, and showed its potential in the regulation of corresponding genes and associated functions. To show the role of cPEP, authors have covered crucial plant processes starting from development to stress response. These findings revealed the possibility of modulating the expression of any coding gene by external application of synthetic peptides. The study is novel as the role of small peptides in enhancing the translation of their own ORFs has not been reported as yet. The study will not only interest groups working in the area of plant science but also open a new area on the regulatory aspect of gene expression across biological sciences. The identification and use of key cPEPs can enhance crop yield and tolerance towards various stresses as well as address weed growth management as an alternative approach to pesticides and chemicals.

Though the manuscript content is novel and needs timely publication, authors need to perform additional experiments to provide more evidence to support their conclusions. My comments are as follows.

Major comments

1. What is the criteria for selecting cPEPs for a gene? Though authors have shown one example of Luciferase, but it is not much clear. The authors should discuss and explain it in detail.

It seems that there are no particular criteria to design a cPEP. Indeed, as shown in Fig. 1d (see below and extended data Fig. 1 for precise sequence chosen) all the different cPEPs tested are active. Moreover, we added the following paragraph in the discussion:

“The design of cPEPs seems to follow no particular rule, indeed, we designed many peptides targeting the luciferase, and all of them were active to increase luciferase activity. We designed them without any a priori, except the fact that the peptides must be at least a little hydrophilic, in order to facilitate their solubilization. In parallel, all the peptides we tested for their molecular activity led to an increased expression of their targeted protein, suggesting that there is no rule to design cPEPs.”

New Fig 1d

2. To strengthen the hypothesis and results, authors should use Arabidopsis mutants of a few genes to show that cPEP treatment does not have any effect on mutants.

The reviewer is totally right. To answer his/her question, we now show that two mutants for genes of two different species (*M. truncatula* and *A. thaliana*) are insensitive to cPEPs (new Fig. 2g, new extended data Fig. 2).

New Fig 2g

New Extended data Fig. 2

3. Authors need to come out with some hypotheses in relation to the action of cPEPs to enhance the translation of mRNAs. Authors should make efforts to prove the proposed hypothesis.

As suggested by the reviewer, we now show in the revised version of our manuscript that cPEPs activate translation of their targeted protein (new Fig. 4a) since cPEPs are not able to activate translation when plants are also treated with cycloheximide (CHX), a translation inhibitor.

New Fig. 4a

Moreover, a 5'Pseq approach was performed in order to reveal ribosome dynamics (Pelechano et al., 2015 Cell) after cPEP treatment. The analysis revealed that reads are more abundant in the 5' region surrounding ATG codon after cPEP treatment. These data strongly suggest that cPEP treatment induces recruitment of ribosomes to enhance translation of its target (new Fig. 4c).

New Fig. 4c

Finally, we show that cPEPs can be detected in polysomal fractions supporting its role in translation regulation (Fig. 4f), and that cPEPs can interact with ribosomal protein (extended data Fig. 5).

New Fig. 4f

4. The authors have only two figures in the main document. As the study has novelty, authors need to divide figures (at least into 4) with detailed legends so that readers can understand the novelty of the work.

The new version of the manuscript has now 6 main figures and 5 extended data figures, as well as 3 extended data tables. We have tried to make effort concerning the legends to allow readers to better understand our work.

5. On the basis of the work done, authors need to make a model for the hypothesis and use it as a figure.

As suggested by the reviewer, we have now included a model representing the mode-of-action of cPEPs in plant cells (new Fig. 6).

New Fig. 6

Minor Comments

1. In the figure 1 legend, the authors have used the term irrelevant peptide and artificial peptide. What do these terms exactly stand for? Please elaborate on it for better understanding.

There was a misunderstanding with these terms, we have now only kept the term “Irrelevant Peptide” to designate a 10 amino acid-peptide carrying no homology with a plant genome (See extended data Table 1), and that we used as a control in all the experiments in which we screened many peptides.

2. In one experiment, the authors used a mixture of different peptides. Whether the authors used each peptide with the same concentration or the concentration of the mixture was taken into the consideration.

It is now written in online Methods: “For growth experiments, *Barbarea vulgaris*, *A. hypochondriacus*, *A. thaliana* and *G. max* seedlings were treated just after sowing and 3 times a week with 500 μ L of a mix of 20 μ M of each peptide.”

3. In line no. 94, there is a typo error in fig. 4i, j it should be corrected Fig2i, j

Thanks for the advices, it has been corrected in the new version of the manuscript

4. Add at least one reference in the statement mentioned in lines 27-33.

According to the reviewer’s remark, we added two references :

3. Duke SO, Dayan FE. The search for new herbicide mechanisms of action: Is there a 'holy grail'? *Pest Manag Sci.* **78**:1303-1313 (2022).

4. Myers JP et al. Concerns over use of glyphosate-based herbicides and risks associated with exposures: a consensus statement. *Environ Health*. **15**:19 (2016).

5. Regarding statistical analysis in fig 2a, the lesion area of cpk3 seems more significant; please re-check it.

We show in the new Fig. 3e that the lesion area of plants treated with cPEPcpk3 is significantly different from the control, according to a Student t-test, with a p-value < 0.05.

Fig. 3e

Fig. 3e with a Tukey comparison

According to the reviewer suggestion, we still performed an ANOVA with a Tukey comparison (see figure above) showing that the lesion area of plants treated with cPEPcpk3 is more significantly different from the control than other cPEPs. However, in a wish of homogeneity between figures, and because this information did not appeared relevant enough for the main message of the manuscript, we chose to let stars to show a significant difference with the control.

Reviewer #2 (Remarks to the Author):

The paper of Ormaney et al., was initiated by the earlier work of these authors concerning influence of miPEPs on the transcription of pri-miRNAs. It has been shown that miPEPs act as transcriptional activators responding to the cis-elements which represent sequences of the synthesized RNA chains including the own coding ORFs. Undoubtedly, this is pioneering study of great impact. However, the present work makes a twofold impression.

Evidently, biotechnological perspectives of the observed phenomenon could be substantial. On the other hand, fundamental basis of the research is weakly supported. First, the authors show that the effect of synthetic cPEPs targeting luciferase gene in all 3 ORFs and specifically capable to interact with luciferase mRNA is not connected with mRNA abundance (i.e. transcription). This is in contradiction with the published data of these authors obtained on pri-miRNA+miPEP model.

I feel this contradiction requires comprehensive set of experiments to support understanding the potential mechanisms of action of cPEPs. First, the experiments should be done to discriminate between nucleus-localized and cytoplasm-localized mechanisms. Particularly, in nucleus there can be effect on RNA processing (capping and polyadenylation) as well as on mRNA transfer through nuclear envelope to cytoplasm. All the above processes may influence the amount of synthesized enzyme without increasing the total mRNA concentration. In cytoplasm, the exogenic cPEP may bind to its coding mRNA sequence and somehow may expedite translation. Particularly, this binding could influence the structure of the cytoplasmic luciferase-specific non-translatable mRNPs and makes probable dormant mRNPs more accessible for ribosomes.

Anyway, the additional experiments should be done irrespective actual mechanism of the cPEP action, and this paper cannot be accepted in its present form.

This remark is really pertinent. Our new data do not contradict the miPEP mechanism. In fact, these are two different mechanisms, depending on the location of the interaction between the peptide and RNA: concerning miPEPs, their localization was shown to be in the nuclei and the interaction revealed by FRET FLIM experiments with their nascent pri-miRNA occurs in nucleus (Lauressergues et al., 2022). Moreover, our preliminary data suggest that miPEPs can interact with some members of the Dicing complex localized in nuclei which have been shown by others to be involved in pri-miRNA transcription, These results are consistent with the functions of miPEP in activating microRNA gene transcription.

Concerning the cPEPs, the situation is different. Our new data reveal that cPEPs activate translation and their interaction with RNA occurs in the cytoplasm and not in the nucleus, as revealed by our FRET FLIM experiment (new fig. 2a).

New Fig. 2a

Besides, we also show that cPEPs increase the translation of their targeted protein (new Fig. 4a). As you can see below, a treatment of plants with the translation inhibitor cycloheximide (CHX) leads to an incapacity of cPEPs to modulate protein translation.

New Fig. 4a

Moreover, a 5'Pseq approach was performed in order to reveal ribosome dynamics (Pelechano et al., 2015 Cell) after cPEP treatment. The analysis revealed that reads are more abundant in the 5' region surrounding ATG codon after cPEP treatment. These data strongly suggest that cPEP treatment induces recruitment of ribosomes to enhance translation of its target (new Fig. 4c).

New Fig. 4c

Finally, we show that cPEPs can be detected in polysomal fractions supporting its role in translation regulation (Fig. 4f), and that cPEPs can interact with ribosomal protein (extended data Fig. 5).

New Fig. 4f

Reviewer #3 (Remarks to the Author):

This study firstly reported the cPEP could induce the relative protein abundance. cPEP_{LUC} can specifically target LUC proteins and increase the abundance of the targeted protein, and the authors have already bear out the best maximal effect of cPEP treatment included concentration, time and amino acid length. Based on this results, there were more proteins were chosen, and the relative cPEP can also induce target protein abundance. It means that cPEP is broad-spectrum. In order to verify the specific functions of these cPEPs, they proved that the increased protein abundance after cPEP treatment were sufficient to modulate plant development. Such as enhanced plant defense against the necrotrophic pathogen, decreased chlorophyll content, decreased flowering days and leaf growth. All data showing a synergistic effect of cPEPs, and it is significant to regulate several plant phenotypes.

I think this finding is interesting and informative for the plant/crops grow regulation. The external application of synthetic peptides cPEP can be used for agronomic purposes to improve crop yield and decrease weed growth. And these types of peptides represent a credible alternative to chemicals. This report gave us a new method to face the biotic and abiotic stresses, it also contributed to the crop breeding and cultivation. While the following suggestions should be considered:

Q1: In this work, the authors listed the predicted small peptides which encoded by 3 ORF. For LUC protein, they selected several parts of the sequence, and all of the sequence works well. But for the growth or stress response genes, only one sequence was selected to carry out the experiment, what is the criteria for selecting the cPEP sequence for the endogenous genes? If the ORF2 encoded small peptides were selected to treat the plant, will it get similar results? Or the results will be same as the LUC protein.

We have seen on luciferase (and we have checked it on two other genes, nsp1 and GFP, data not shown, see figure below) that it seems that there is no particular rule/condition to design cPEP (as we added in our new results and discussion). This is the reason why we only took one cPEP per gene in the next experiments, independently of the reading frame. Moreover, all the different proteins tested both by Western blot and phenotyping (SKL, BAK1, BRI1, CPK3, DCL1, EIN2, HSP101) gave positive results (Fig. 2h, 3a, 3d, 3e, 3f and extended data table 3).

Data not shown: effect of different cPEPs designed on GFP (left) and NSP1 right) sequence, in the three different frames

Q2: the cPEP sequences of the 12 endogenous genes should be supplied in the supplementary data.

All the sequences of cPEPs are now provided in Extended Data Table 1

Q3: Have you considered the poor specificity of 5-10 amino acids among all proteins in plants? Since they may induce other protein abundance. And considering the diversity of protein and amino acid between different species, the cPEP of model plant has the same effect in other plants? As I see, when we did research on circRNA, we overexpressed the linear sequence of circRNA, sometimes it also got a phenotype of induced stress resistance (but not all of the linear sequence can increase stress resistance). Thus, do you think it can be

explained by your report? Did you find other endogenous genes which protein abundance can not be induced by cPEP?

Thank you for this relevant question. To answer it, we performed a proteomic analysis of plants treated by a cPEP compared to plants treated with a scrambled peptide. We chose the luciferase gene as a target to avoid the effect of cPEPs on a full signalling pathway by targeting a transcription factor or a kinase. The new Fig. 1h shows that treatment with a cPEP only targets the targeted protein, and has no effect on the full proteome of a plant.

Fig. 1h

Q4: In Fig. 1 b-e, you use “Scrambled Cpep”, but in Fig. 2 you use “Irrelevant Peptide”. So What is the different between “Scrambled” and “Irrelevant”? And there are some writing mistake in the figures for the word “Irrelevant”, for example, Fig 1c,d.

We thank the reviewer for pointing these errors, we had a look deeper to remove several mistakes. The sequences of all the peptides used are listed in Extended Data Table 1. We mainly used a scrambled peptide corresponding to the tested cPEPs (same amino acid composition but sequence order is different). But, in some cases, as when we screened different peptides for a determined phenotype, we used an irrelevant peptide, i.e. a peptide without any homology in a plant genome, as a control for our experiments.

Q5: It is recommended that pictures in Fig. 2 cd, ef, ig and Extended data Fig. 2 gh, ig be arranged horizontally in the same order as another picture.

Creating figures when the pictures are not in the same orientation might be difficult. We tried to modify as much as possible our new figures to take into account this remark

Reviewer #4 (Remarks to the Author):

In this manuscript the authors demonstrate that a peptide with a sequence which is complementary to that of a protein can result in enhanced translation of the target protein. The results are convincing, but a mechanistic model explaining the action of these cPEP is missing. The authors make reference to their previous evidence that miPEP derived from ORFs in miRNA gene can lead to enhanced transcription of the corresponding miRNA, but here the mechanisms appear to be different, being transcription of the cPEP-target gene unaffected. Interaction of the cPEP with the LUC mRNA is shown, but how is this related to enhanced translation? How does the cPEP interact with its corresponding mRNA? All these questions are not even addressed in the manuscripts.

The reviewer is totally right, in the new version of our manuscript, we added new experiments revealing that cPEPs increase translation of targeted protein (new Fig. 4a and 4c, new Extended Data Figure 3) by interacting with their own nascent RNA (new Fig. 2a) and ribosomal proteins (new Fig. 4f and Extended Data Fig. 5).

First, by using the translation inhibitor cycloheximide (CHX), we now show that cPEPs can't activate translation of their target proteins when plants are exogenously treated with CHX (new Fig. 4a).

New Fig. 4a

Moreover, a 5'Pseq approach was performed in order to reveal ribosome dynamics (Pelechano et al., 2015 Cell) after cPEP treatment. The analysis revealed that reads are more abundant in the 5' region surrounding ATG codon after cPEP treatment. These data strongly suggest that cPEP treatment induces recruitment of ribosomes to enhance translation of its target (new Fig. 4c).

New Fig. 4c

Finally, we show that cPEPs can be detected in polysomal fractions supporting its role in translation regulation (Fig. 4f), and that cPEPs can interact with ribosomal protein (extended data Fig. 5).

New Fig. 4f

Finally, we also added a new Fig. 6 where we present the mode-of-action of cPEPs in plants.

I would suggest the authors to utilize box-plots instead of histograms to represent their data. Why is SEM used instead of SD when doing the statistical analysis?

We thank the reviewer for its suggestion. Histograms (bar plots) and box plots are very similar in that way they are both used to visualize numeric datasets. According to the reviewer's suggestion, we have modified the figure 5 to add box plots (see below, new Fig. 5). However, we are not really convinced that, for our data, this representation would be more relevant.

We preferred to use SEM since the meaning of SEM includes statistical inference based on the sampling distribution. SEM is the SD of the theoretical distribution of the sample means (the sampling distribution).

Fig. 5

First version of Fig. 5

Fig. 5

New Fig. 5

Reviewers' Comments:

Reviewer #1:

Remarks to the Author:

I have gone through the revised manuscript and the responses of the authors. The authors have revised the manuscript as per the suggestions. The manuscript can be accepted for publication.

Reviewer #2:

Remarks to the Author:

This manuscript has been substantially improved and can be accepted as it is.

Reviewer #3:

Remarks to the Author:

I appreciate the efforts made by the authors to answer the requests of clarification and revision. They revised all of the mistakes in the manuscript as suggested, and added several experiment to verify the mechanisms how cPEPs active the target proteins accumulation and then induce the relative phenotypes of plant. The whole manuscript looks well now. I do not have any other questions.

Reviewer #4:

Remarks to the Author:

The revised version of this manuscript is quite improved vs. the original one. The authors now provide convincing evidences that cPEPs control their "partner" mRNA translation. All the experimental work is focused on the use of exogenous cPEPs, but the existence of a similar process in vivo is intriguing and would require additional work. Is it possible the cPEPs originate in vivo from the degradation (turn-over) of the corresponding proteins? This is something for a follow-up work, but the authors may want to add some more speculation on the biological significance of the cPEPs in vivo.

Minor comment: In the new version of In the legend of Fig.2 we read: "wild type or NSP1 version in which the cPEP-encoding sequence was removed (NSP1 Δ cPEP)". This is probably not correct, since the sequence that was removed was not actually "encoding" for the cPEP. If cPEPs are encoded by the mRNA of the cPEP regulated gene is unknown. I would rather define this sequence as the mRNA sequence corresponding to the cPEP sequence (or "matching the cPEP"). I also suggest the authors to provide a clearer explanation of the corresponding experiment in Fig 2, because I found it not straight forward.

REVIEWER COMMENTS

Reviewer #1 (Remarks to the Author):

I have gone through the revised manuscript and the responses of the authors. The authors have revised the manuscript as per the suggestions. The manuscript can be accepted for publication.

Reviewer #2 (Remarks to the Author):

This manuscript has been substantially improved and can be accepted as it is.

Reviewer #3 (Remarks to the Author):

I appreciate the efforts made by the authors to answer the requests of clarification and revision. They revised all of the mistakes in the manuscript as suggested, and added several experiment to verify the mechanisms how cPEPs active the target proteins accumulation and then induce the relative phenotypes of plant. The whole manuscript looks well now. I do not have any other questions.

We would like to thank the reviewers for the helpful comments which allowed us to improve the quality of our manuscript

Reviewer #4 (Remarks to the Author):

The revised version of this manuscript is quite improved vs. the original one. The authors now provide convincing evidences that cPEPs control their "partner" mRNA translation. All the experimental work is focused on the use of exogenous cPEPs, but the existence of a similar process in vivo is intriguing and would require additional work. Is it possible the cPEPs originate in vivo from the degradation (turn-over) of the corresponding proteins? This is something for a follow-up work, but the authors may want to add some more speculation on the biological significance of the cPEPs in vivo.

We would like to thank the reviewer for this comment. As requested, we added in the new version of the discussion of the manuscript a paragraph discussing the possibility and the potential sources of natural cPEPs:

“Here we described the use of cPEPs to externally modulate the translation of transcripts coding for proteins (Fig. 6). All tested peptides were designed artificially, with the use of bioinformatics, and we cannot exclude that cPEPs exist *in planta*. Whether such peptides hidden in the plant genomes exist still remains to be determined and constitutes the next line of research. Several recent findings from other groups demonstrate that many short open reading frames hidden in intergenic sequences or within coding sequences have the ability to produce small functional peptides, possibly natural cPEPs (“natcPEPs”), both in plants and other

species¹⁴⁻¹⁶. Whether these natcPEPs impinge on the expression of their target protein remains to be determined.

Several natural sources of natcPEPs can be considered. First, long non-coding RNA (lncRNAs), which have already been shown to encode small peptides^{14,15}. A second source of natcPEPs would be the coding genes themselves. Indeed, 5' and 3' "Untranslated Regions (UTRs)" as well as alternative ORFs (named altprot) present in the main ORFs of coding genes can encode small peptides¹⁴⁻¹⁶. Finally, another potential source of natcPEPs might be peptides resulting from protein degradation, produced by the proteasome or other intracellular proteases. An enticing hypothesis would be that short peptides produced by proteases could act as cPEPs in order to compensate degradation to maintain a steady state level of cPEP-targeted proteins. The main issue in identifying natcPEP sources comes from the difficulty to detect these peptides *in planta*. Indeed, the identification of such small molecules by mass spectrometry still remains challenging¹⁷."

Minor comment: In the new version of In the legend of Fig.2 we read: "wild type or NSP1 version in which the cPEP-encoding sequence was removed (NSP1 Δ cPEP)". This is probably not correct, since the sequence that was removed was not actually "encoding" for the cPEP. If cPEPs are encoded by the mRNA of the cPEP regulated gene is unknown. I would rather define this sequence as the mRNA sequence corresponding to the cPEP sequence (or "matching the cPEP"). I also suggest the authors to provide a clearer explanation of the corresponding experiment in Fig 2, because I found it not straight forward.

Many thanks for this right comment, we corrected "encoding" by "corresponding", and tried to modify slightly the result part and the Fig 2 legend to render the story clearer.

Reviewers' Comments:

Reviewer #4:

Remarks to the Author:

The authors have replied to all my comments. This is an exciting discovery. I have non further concerns.